# Global Analysis of microRNA-like RNAs Reveals Differential Regulation of Pathogenicity and Development in *Fusarium oxysporum HS2* Causing Apple Replant Disease

**DOI:** 10.3390/jof10120883

**Published:** 2024-12-19

**Authors:** Ruxin Zhao, Xiangmin Suo, Xianglong Meng, Yanan Wang, Pengbo Dai, Tongle Hu, Keqiang Cao, Shutong Wang, Bo Li

**Affiliations:** 1State Key Laboratory of North China Crop Improvement and Regulation, College of Plant Protection, Hebei Agricultural University, Baoding 071000, China; 16632235120@163.com (R.Z.); cugmxl@163.com (X.M.); wyn3215347@163.com (Y.W.); daipengbo@hebau.edu.cn (P.D.); tonglemail@126.com (T.H.); cao_keqiang@163.com (K.C.); 2Shijiazhuang Institute of Fruit, Hebei Academy of Agriculture and Forestry Sciences, Shijiazhuang 050051, China; suoxiangmin@126.com; 3Institute of Agricultural Information and Economics, Hebei Academy of Agriculture and Forestry Sciences, Shijiazhuang 050051, China

**Keywords:** *Fusarium oxysporum*, apple replant disease, microRNA-like RNAs, sRNA-seq, stage-specific regulation

## Abstract

This study investigated the expression profiles of microRNA-like RNAs (milRNAs) in *Fusarium oxysporum HS2* (*FoHS2*), a key pathogen causing Apple replant disease (ARD), across spore to mycelium formation stages. Using small RNA sequencing (sRNA-seq) and bioinformatics, we identified and analyzed milRNAs, revealing their targeting of 2364 mRNAs involved in 20 functional categories, including metabolic and cellular processes, based on gene ontology (GO) analysis. An analysis of Kyoto Encyclopedia of Genes and Genomes (KEGG) showed that these mRNAs are related to carbohydrate and amino acid metabolism pathways. Notably, the highest number of differentially or specifically expressed milRNAs (DEmilRNAs/SEmilRNAs) was found during the spore stage, with *FoHS2*-milR19 targeting genes encoding histone acetyltransferases, methyltransferases, and cell wall-degrading enzymes (CWDEs), which are crucial for growth, development, and pathogenicity. We validated the reliability of our sRNA-seq data and the expression of target genes using stem-loop RT-PCR and qRT-PCR. Our results highlight the stage-specific expression of milRNAs in *FoHS2*, particularly in the spore stage, suggesting a key role in regulating host life activities and providing a theoretical basis for developing RNA-based pesticides to control ARD.

## 1. Introduction

MicroRNAs (miRNAs), a type of endogenous non-coding single-stranded RNA of 18–24 nucleotides, play roles in various biological processes. MiRNA regulates gene expression negatively through complementary or partially complementary binding to target mRNA, leading either to mRNA degradation or translational inhibition, thus playing essential roles in cell differentiation, growth, development, and stress response [1]. Since the discovery of the miRNA gene *lin-4* in *Caenorhabditis elegans* in 1993 [2], research on miRNA has rapidly expanded to both animals and plants, revealing the conservation and diversity of miRNA across different species [3,4,5,6]. However, the existence of miRNAs in fungi has long been a matter of debate. The traditional view is that miRNAs predominantly exists in multicellular eukaryotes, whereas fungi, with simpler structures, might lack such complex gene regulatory mechanisms. In 2010, researchers identified miRNA-like RNAs (milRNAs) in *Neurospora crassa*, challenging this view and confirming the presence of milRNAs in fungi [7]. The biogenesis of milRNAs involves a set of proteins similar to those in plants and animals, including Dicers, exonucleases like QIP, AGO proteins such as QDE2, and proteins with RNase III domains, such as MRPL3. These milRNAs are transcribed by RNA polymerase II and are capable of forming protruding, self-complementary stem-loop structures [8], which are crucial for their function in mediating gene silencing at both the transcriptional and post-transcriptional levels [9]. Subsequently, milRNAs have been identified in a variety of other fungi, including *Sclerotinia sclerotiorum* [10], *Trichoderma reesei* [11], *Curvularia lunata* [12], *Trichophyton rubrum* [13], *Puccinia triticina* [14], *Coprinopsis cinerea* [15], *Botryosphaeria dothidea* [16], *Rhizoctonia solani* [17], and *Fusarium verticillioides* [18]. These studies have enriched our understanding of milRNA diversity in fungi and revealed the critical role of milRNAs in fungal lifecycles. For instance, milRNAs in *S. sclerotiorum* are involved in the regulation of sclerotial development [19], whereas in *Aspergillus oligospora*, milRNAs are closely related to the transition of lifestyles [20]. Moreover, milRNAs in *F. oxysporum* have also shown significant biological functions. For example, milRNAs in *F. oxysporum* f. sp. *cubense* promote the infection of bananas by suppressing virulence inhibitors [21]. *Fol*-milR1 in *Fol* enhances pathogenicity by silencing host resistance genes, thus weakening the tomato immune system [22]. These findings emphasize the central role of milRNAs in the physiological development of fungi and in the interaction between pathogenic fungi and their host plants.

Apple replant disease (ARD) is a typical soil-borne disease of woody plants caused by *F. oxysporum* and has become a major obstacle restricting the development of the global apple industry. This disease, which is characterized by its ability to infect numerous economically important crops including fruit trees, vegetables, and cereals, disrupts root development and negatively impacts fruit yield and quality. Compared to the infection of herbaceous plants, the infection of woody plants by *F. oxysporum* has notable particularities. Firstly, the growth cycle of woody plants is longer, and the infection latency period of *F. oxysporum* can last for several months, presenting a considerable challenge for the early diagnosis and control of diseases. This disease impedes the normal growth and development of apple tree roots, reducing both fruit production and quality, as well as severely impacting propagation efficiency in nurseries and the long-term sustainability of orchards [23,24,25,26]. Despite progress in research on milRNAs in *F. oxysporum*, the majority of studies have concentrated on herbaceous plant pathogens, overlooking the role of milRNAs in the infection of woody plants by this fungus. Specifically, the investigation of milRNAs in *F. oxysporum* in the context of ARD remains largely unexplored. As a typical vascular bundle pathogen, ARD has unique biological characteristics in its infection process. Due to the complexity of the infection process of *F. oxysporum* in apple trees, involving a long incubation period and gradual destruction of the vascular system, the production and germination of conidia are one of the key factors for successful infection. Thus, understanding the role of milRNAs in this process is particularly important. Exploring the types, expression patterns, and associations of milRNAs with pathogenesis in *F. oxysporum* can advance our understanding of disease mechanisms of ARD and help develop novel control strategies.

In this study, we aim to comprehensively investigate the composition, expression changes, and structural characteristics of milRNAs in the highly pathogenic *F. oxysporum HS2* strain (*FoHS2*), isolated and identified in our prior research [27], across different growth stages using sRNA-seq. Meanwhile, bioinformatics approaches were employed to examine the potential regulatory roles of stage-specific milRNAs and their target genes in the growth, development, and pathogenicity of the *FoHS2*. This research intends to identify key milRNAs across different growth stages, offering insights into the interaction mechanisms between *F. oxysporum* and apple trees and providing a scientific foundation for developing effective control measures for ARD.

## 2. Materials and Methods

### 2.1. Fungal Strains and Sample Preparation

The *FoHS2* strain was obtained from the Laboratory of Plant Disease Epidemiology and Integrated Control at Hebei Agricultural University. For the strain *FoHS2*, isolated from *Malus domestica* roots in Hengshui, Hebei Province in 2011, identification encompassed morphological analysis and molecular methods. *FoHS2* was cultured on potato dextrose agar (PDA) medium (containing 200 g peeled potato, 20 g glucose, and 18 g agar per liter) at 25 °C in the darkness for five days. Agar plugs (0.8 cm in diameter) were then collected from the colony edges, with 3–5 plugs transferred to 100 mL of potato dextrose broth (PDB, containing 200 g of peeled potato and 20 g glucose per liter), and incubated in a shaker at 25 °C and 150 rpm for eight days. The spore suspension was filtered using sterile gauze and transferred to a 50 mL RNase-free centrifuge tube followed by centrifugation at 4 °C, 3000 rpm to collect the spore precipitate in triplicate. One portion of the pellet sample was snap-frozen in liquid nitrogen and stored at −80 °C for library preparation of sRNA-seq as the spore sample (*Fo*_sp). The remaining two spore samples were each cultured in 100 mL of yeast extract peptone dextrose (YEPD, containing 10 g yeast extract, 10 g peptone, and 20 g glucose per liter) at 25°C and 150 rpm for 12 and 48 h, respectively, to obtain germinating spores (*Fo*_gs; in this study, the germination stage was defined as the number of spores with germ tube length ≥ twice the shortest germ tube diameter that reached 60%) and mycelium (*Fo*_myc) samples. These samples were collected via centrifugation (4 °C, 3000 rpm), with germinating spores and mycelium transferred to RNase-free tubes, snap-frozen in liquid nitrogen, and stored at −80 °C for subsequent sequencing library construction. Samples from the three growth stages were designated as *Fo*_sp, *Fo*_gs, and *Fo*_myc (Appendix A), with biological replicates *Fo*_sp-1, *Fo*_sp-2, and *Fo*_sp-3; *Fo*_gs-1, *Fo*_gs-2, and *Fo*_gs-3; and *Fo*_myc-1, *Fo*_myc-2, and *Fo*_myc-3.

### 2.2. Construction and Sequencing of sRNA Libraries

TransZol Up (Transgenic Biotech, Beijing, China) and the QIAseq miRNA library kit (Qiagen Inc., Valencia, CA, USA) were employed to extract total RNA from tissue samples of *Fo*_sp, *Fo*_gs and *Fo*_myc and to construct libraries, respectively. The concentration and purity of the isolated RNA were assessed using the Nanodrop 2000 (Thermo Scientific Inc., Waltham, MA, USA), and the integrity of the RNA was verified via agarose gel electrophoresis. RNA quality number (RQN) was measured with the Agilent 5300 (Agilent Technologies Inc., Santa Clara, CA, USA) to ensure the RNA met the library construction criteria (OD260/280 = 1.8–2.2, OD260/230 > 2.0, 28S: 18S > 1.0). Adapter sequences were ligated to both the 3′ and 5′ ends, followed by reverse transcription to synthesize cDNA for library enrichment and purification. Sequencing was performed on the Illumina NovaSeq X Plus platform (Majorbio, Shanghai, China) at Shanghai Majorbio Bio-pharm Biotechnology Co., Ltd. Raw data were processed to remove adapter sequences and low-quality reads (quality score < 20, filtered out reads containing ambiguous nucleotides (N), or reads < 18 nt or > 32 nt in length) to ensure high-quality data for subsequent analysis (https://github.com/agordon/fastx_toolkit, accessed on 25 December 2023). Each library included three biological replicates.

### 2.3. Identification of milRNAs in FoHS2

The quality-controlled useful reads were aligned to the reference genome (JARFYI000000000) using Bowtie2 (http://bowtie-bio.sourceforge.net/bowtie2/index.shtml, accessed on 25 December 2023) to obtain mapped reads for subsequent prediction, identification, and expression quantification of milRNAs in *FoHS2*. The mapped reads first compared with mature and precursor miRNA sequences in the miRBase 22.0 database (http://www.mirbase.org/, accessed on 25 December 2023), allowing for a maximum of one nucleotide mismatch; reads that matched were considered as known miRNAs. Unmatched reads were further aligned to the reference genome, and surrounding sequences were analyzed with miRDeep2 (https://www.mdc-berlin.de/content/mirdeep2-documentation, accessed on 25 December 2023) for secondary structure prediction and Dicer processing analysis, identifying novel miRNAs based on characteristic hairpin structures.

### 2.4. Expression Analysis of milRNAs at Three Different Growth Stages

Quantitative analysis of miRNAs expression levels was conducted using DESeq2 (http://bioconductor.org/packages/stats/bioc/DESeq2/, accessed on 25 December 2023), with transcripts per million (TPM) normalization applied for cross-sample comparisons. DEmilRNAs between *Fo*_sp, *Fo*_gs, and *Fo*_myc samples were identified using false discovery rate (FDR) < 0.05 and |log_2_ Fold Change (FC)| ≥ 1.

### 2.5. Verification of the Expression Level of milRNAs via Stem Loop qRT-PCR

Two highly expressed milRNAs, *FoHS2*-milR1 and *FoHS2*-milR2, were selected for validation via stem-loop RT-qPCR. Stem-loop primers, specific forward primers and universal reverse primers were designed using the miRNA Design V1.01 from Vazyme (Nanjing, China) and synthesized by Sangon Biotech Co., Ltd., which is located in Shanghai, China. (Appendix A). The total RNA of *Fo*_sp, *Fo*_gs, and *Fo*_myc samples was extracted with TransZol Up (TransGen Biotech, Beijing, China), and RNA concentration was assessed with a NanoDrop (Thermo Scientific Inc., Waltham, MA, USA). One reverse transcription of total RNA was performed using stem-loop primers according to the instructions of the miRNA 1st Strand cDNA synthesis kit (by stem-loop) (Vazyme, Nanjing, China), and the resulting cDNA was used as a template for qRT-PCR. qRT-PCR reactions included SYBR green dye (BestEnzymes, Lianyungang China) and a snRNA U6 as an internal reference. Relative expression levels were calculated using the 2^−ΔΔCT^ method and visualized with GraphPad Prism 9.0.

### 2.6. Target Gene Identification and Functional Analysis

MilRNA target mRNAs were predicted using RNAhybrid (http://bibiserv.techfak.uni-bielefeld.de/rnahybrid/, accessed on 25 December 2023), psRobot (http://omicslab.genetics.ac.cn/psRobot/index.php, accessed on 25 December 2023), and TargetFinder (https://github.com/carringtonlab/TargetFinder, accessed on 25 December 2023) and aligned to GO (http://geneontology.org/, accessed on 25 December 2023) and KEGG database (https://www.kegg.jp/, accessed on 25 December 2023) for functional annotation. GO and KEGG enrichment analysis were performed using Goatools (https://github.com/tanghaibao/GOatools, accessed on 25 December 2023) and the scipy Python package (https://scipy.org/install/, accessed on 25 December 2023), with Fisher’s exact test and Benjamani–Hochberg (BH) multiple test correction (*p* < 0.05). The sRNA-seq data have been deposited in the SRA database under BioProject ID: PRJNA1159741 and submission ID: SUB14694852.

## 3. Results

### 3.1. Small RNA Sequencing and Data Analysis

To identify the milRNAs in *FoHS2*, we constructed separate cDNA libraries for three growth stages: spore (*Fo*_sp), germinated spore (*Fo*_gs), and mycelium (*Fo*_myc). Sequencing yielded an average of 6,225,626, 6,044,342, and 3,193,344 clean reads for *Fo*_sp, *Fo*_gs, and *Fo*_myc, respectively. Following quality control, we obtained average useful reads of 5,945,197, 5,677,506, and 3,143,721, with useful reads comprising over 90% of the total reads for each sample set (Table 1 and Appendix A). These results indicate high-quality sRNA-seq data suitable for subsequent analysis.

In-depth analysis of the sequencing data identified 127, 70, and 59 milRNAs in *Fo*_sp, *Fo*_gs, and *Fo*_myc, respectively (Appendix A). The identified milRNAs ranged in length, from 18 to 24 nucleotides, with 21-nucleotide milRNAs being the most prevalent (Figure 1a–c). Additionally, the first nucleotide of these milRNAs exhibited a strong preference for uracil (U), with U as the initial base in 53.54%, 63.24%, and 70.69% of the milRNAs in *Fo*_sp, *Fo*_gs, and *Fo*_myc, respectively (Figure 1d–f). This bias towards uracil may be attributed to the selectivity of the Argonaute protein, which exhibits a strong predilection for 5′-uracil when choosing the guide strand of the miRNA duplex [28]. To provide a clear overview of expression patterns, we present detailed information on the top 10 most highly expressed milRNAs in *Fo*_sp, *Fo*_gs, and *Fo*_myc, including transcripts per million (TPM) values and sequence information (Appendix A). Furthermore, we performed hairpin structure prediction and analysis for precursor sequences of the core milRNAs (Appendix A). These hairpin structures, a critical feature of milRNA biosynthesis, further confirmed the authenticity and reliability of the candidate milRNAs.

### 3.2. Expression Analysis of milRNAs in Three Growth Stages of FoHS2

In this study, we used a pairwise comparison approach to investigate the expression profiles of milRNAs across three distinct growth stages: *Fo*_sp, *Fo*_gs, and *Fo*_myc. The analysis revealed that twenty-six milRNAs were commonly expressed across all three libraries, with ninety-four, seventeen, and three milRNAs specifically expressed in *Fo*_sp, *Fo*_gs, and *Fo*_myc, respectively. Notably, the *Fo*_sp stage harbored the largest number of SEmilRNAs (Figure 2a). To distinguish the expression characteristics of milRNAs across these stages, we analyzed their expression patterns. During the *Fo*_sp stage, comparison with *Fo*_myc identified 26 DEmilRNAs, with 10 upregulated and 16 downregulated. Among them, *FoHS2*-milR14 showed the highest upregulation (log_2_FC = 7.170561984, *p* = 2.22 × 10^−7^), while *FoHS2*-milR13 exhibited the most significant downregulation (log_2_FC = −6.819607046, *p* = 2.11 × 10^−6^). When *Fo*_sp was compared to *Fo*_gs, eighteen DEmilRNAs were identified, including six upregulated and twelve downregulated milRNAs, with the highest upregulation observed in *FoHS2*-milR14 (log_2_FC = 6.115211663, *p* = 1.03 × 10^−6^) and the highest downregulation in *FoHS2*-milR48, *FoHS2*-milR44, and *FoHS2*-milR50 (log_2_FC = −7.030723533, *p* = 1.00 × 10^−7^). In the *Fo*_sp stage, core stage-SEmilRNAs included *FoHS2*-milR17, *FoHS2*-milR19, *FoHS2*-milR32, etc. In the *Fo*_gs stage, comparing it with *Fo*_myc revealed fourteen DEmilRNAs, with ten upregulated and four downregulated milRNAs. Among them, *FoHS2*-milR49 showed the highest upregulation (log_2_FC = 5.360907491, *p* = 6.85 × 10^−6^); while *FoHS2*-milR45 exhibited the most significant downregulation (log_2_FC = −4.295889629, *p* = 0.009117857) (Figure 2b, Additional file: Appendix A). During the *Fo*_gs stage, important stage SEmilRNAs included *FoHS2*-milR16, *FoHS2*-milR33, *FoHS2*-milR41, etc. In contrast, only three unique milRNAs were identified in the *Fo*_myc stage: aly-miR829-5p, fve-miR11291, and sbi-miR5568f-3p.

Additionally, the expression levels of commonly expressed milRNAs (CEmilRNAs) across the three stages displayed notable differences in expression levels, indicating distinct stage-specific expression patterns. Specifically, ten milRNAs (including *FoHS2*-milR35, *FoHS2*-milR6, *FoHS2*-milR2, etc.) are highly induced during the spore stage. In contrast, *FoHS2*-milR24, *FoHS2*-milR27, and *FoHS2*-milR26 show significant induction during both spore germination and mycelial stages. Conversely, *FoHS2*-milR4 and *FoHS2*-milR5 are only significantly induced during the mycelial stage (Figure 2c). These findings suggest that different milRNAs may play finely tuned, tissue-specific regulatory roles across the fungal growth stages, with expression patterns closely related to growth phases. Notably, the expression levels of CEmilRNAs (*FoHS2*-milR1 and *FoHS2*-milR2) are the highest among the three growth stages, prompting their selection for quantitative real-time PCR (qRT-PCR) validation. The results confirmed the expression trends observed in the sRNA-seq data across the three growth stages (Figure 2d–g).

### 3.3. Targets Prediction and Functional Analysis of CEmilRNAs in the Three Growth Stages of FoHS2

To elucidate the biological functions of milRNAs in *FoHS2* and the information regarding their regulatory target genes, we initially predicted the target genes of CEmilRNAs across the three growth stages, identifying a total of 2364 target genes (Appendix A and Appendix A). Notably, *FoHS2*-milR20 was predicted to target 1445 mRNAs (Appendix A), while *FoHS2*-milR1 and *FoHS2*-milR2, which were both expressed at the most expression level in all three stages, have only three and fifteen target genes, respectively (Appendix A).

GO annotation revealed that these target genes are mainly associated with 20 functional categories, including key biological processes like cell processing (630 genes), metabolic processes (654 genes), and biological regulation (148 genes). They are also involved in cellular components such as organelles (332 genes), membrane components (474 genes), and cell parts (607 genes), as well as molecular functions like catalytic activity (788 genes), transporter activity (154 genes) and binding (706 genes) (Figure 3a). In addition, these genes were mapped to 20 metabolic pathways via KEGG annotation, such as carbohydrate metabolism (ninety-two genes), amino acid metabolism (sixty-eight genes), lipid metabolism (forty-six genes), and energy metabolism (forty genes). They also played roles in genetic information processes like protein folding, sorting, degradation (fifty-five genes), and translation (sixty-five genes), as well as other pathways including transport and catabolism (thirty-seven genes), signal transduction (six genes), and membrane transport (four genes) (Figure 3b).

Through GO enrichment analysis, we found that the target genes of these CEmilRNAs are primarily enriched in biological processes such as structural molecule activity, cytoplasmic parts, ribosomal subunits, and organic substance biosynthetic processes (Figure 3c). KEGG enrichment analysis revealed that these target genes are mainly enriched in metabolic pathways such as the ribosome, oxidative phosphorylation, and protein export (Figure 3d). These results indicate that milRNAs play a crucial role in multiple biological processes of fungi, including the maintenance of cellular structure and function, metabolic regulation, and genetic information processing, by regulating the expression of their target genes. In addition, *FoHS2*-milR20, which is commonly expressed across the three stages, targets specific genes associated with enzymatic activities: a pectinase gene (contig000006.606) and three mannanase genes (contig000001.2219, contig000001.985, and contig000002.1507), as well as an arabinosidase gene (contig000011.20). Similarly, *FoHS2*-milR29 targets two pectinase genes (contig000001.633 and contig000001.65) and two arabinosidase genes (contig000011.20 and contig000003.125) (Additional file: Appendix A).

### 3.4. Targets Prediction and Functional Analysis of Stage-Specific milRNAs of FoHS2

To clarify the functions of stage-specific milRNAs in *FoHS2*, we predicted the target genes of SEmilRNAs at each growth stages. Results indicated that the stage-specific milRNAs from the spore, germinating spore, and mycelial stages could target 6630, 504, and 76 mRNAs, respectively (Appendix A and Appendix A). Remarkably, the number of target genes varied significantly across different SEmilRNAs. Specifically, *FoHS2*-milR19, which is unique to the spore stage, had the highest number of target genes, targeting a total of 5141 mRNAs. *FoHS2*-milR19 was shown to target key pathogenicity-related genes, including genes encoding acetyltransferases (fourteen genes), methyltransferases (twenty genes), protein kinase genes (sixteen genes), pectin lyase (four genes), and glycosyl hydrolase (twenty-two genes) (Table 2 and Appendix A). These genes are crucial for regulating fungal growth, development, and virulence. Interestingly, transcriptomic data indicated that these target genes show varying degrees of upregulation in the spore germination or hyphal stages where *FoHS2*-milR19 is not expressed (Figure 4). This negative regulatory relationship in expression levels between *FoHS2*-milR19 and its targets further validates their targeting interactions.

Through GO enrichment analysis, we identified that the target genes of SEmilRNAs in the spore stage were predominantly enriched in categories such as cellular components, membrane components, the intrinsic components of membranes, the components of membrane parts, cytoplasmic parts, ribonucleoprotein complexes, ribonucleoprotein complexes in the nucleus, transporter activity, transmembrane transporter activity, ion translocation, and anion translocation (Figure 5a). The target genes of spore germination stage-specific milRNAs were mainly enriched in pre-initiation complex, eukaryotic translation initiation factor 3 complex, eukaryotic 48S pre-initiation complex, eukaryotic 43S pre-initiation complex, reactive oxygen species metabolic process, and cytoplasmic translation initiation complex formation (Figure 5c). During the mycelial stage, the target genes of SEmilRNAs were mainly enriched in telomere organization, telomere maintenance, anatomical structure homeostasis, and helicase activity (Figure 5e).

KEGG enrichment analysis further elucidated that, in the spore stage, SEmilRNAs were prominently associated with the ribosome, fatty acid oxidation, and oxidative phosphorylation pathways (Figure 5b). The target genes of SEmilRNAs in the spore germination stage were enriched in pathways including SNARE interactions in vesicle trafficking, fatty acid oxidation, and tyrosine metabolism (Figure 5d). Mycelial stage-specific milRNA targets were mainly involved in spliceosome formation, non-homologous end joining, and homologous recombination (Figure 5f). Strikingly, KEGG annotation showed that 68 target mRNAs in the spore-specific milRNAs were linked to the secondary metabolite metabolism, of which 47 were targeted by *FoHS2*-milR19 (Appendix A). Moreover, 12 mRNAs targeted by *FoHS2*-milR19 and other spore stage-specific milRNAs were mapped to the MAPK signaling pathway (Appendix A). These results provide an in-depth understanding of the functions of the milRNAs unique to different growth stages of *FoHS2* and reveal their potential roles in regulating fungal growth, development, and pathogenicity.

## 4. Discussion

MilRNAs serve as pivotal regulators of gene expression and play a crucial role throughout the fungal life cycle, particularly in the regulation of mycelium growth, spore formation, and pathogenicity [15,29]. For example, *Vm*-milR1 in the *V. mali* silences two host receptor-like kinase genes, *MdRLKT1* and *MdRLKT2*, to suppress the host’s immune response [30]. To explore the controlling roles of milRNAs in *FoHS2*, a pathogen of woody plants, we first utilized sRNA-seq technology to sequence samples from the three different growth stages of *FoHS2*, identifying 127, 70, and 59 milRNAs in each stage (Appendix A), respectively. Among these, 26 milRNAs were commonly expressed across all three growth stages of the *FoHS2* (Figure 2). We hypothesize that these CEmilRNAs may play regulatory roles throughout the entire life cycle of *FoHS2*. Furthermore, fungal milRNAs can finely regulate their growth, development, and pathogenicity through stage-specific expression pattens. For instance, the milRNAs in the mycelium and spores of *Sphaerosporella brunnea* specifically express and differentially regulate their own growth, development, and reproduction [31]. Similarly, *Cordyceps guangdongensis* possesses a set of conserved and novel miRNAs that exhibit differential expression, indicative of their regulatory role across its developmental spectrum [32].

In this study, the expression patterns of milRNAs at different growth stages of *FoHS2* exhibited stage specificity. For example, *FoHS2*-milR17, *FoHS2*-milR19, and *FoHS2*-milR39 were specifically expressed only during the spore stage (Appendix A). Additionally, we observed that some of these stage-specific milRNAs exhibit relatively low expression levels at particular growth stages, leading us to speculate that the low expression level of SEmilRNAs might be closely related to their unique regulatory functions. Previous studies in plants have established that miRNAs finely tune target gene expression in both temporal and spatial dimensions, which is crucial for their adaptability and survival at different developmental stages and in response to various environmental challenges [33]. Taking this into account, we speculated that in *FoHS2*, certain triggers such as environmental signals, developmental cues, or internal regulatory networks may induce the stage-specific expression of milRNAs, thereby achieving precise control over particular tissues or cell types. Furthermore, target gene prediction analysis revealed that these specifically low-expressing milRNAs often simultaneously regulate multiple target genes, forming a complex regulatory network. This regulation may encompass intricate molecular mechanisms, such as the interplay between miRNAs and their target genes, the trafficking and localization of miRNAs [34], and the synergistic action with other regulatory factors [35]. Therefore, we boldly hypothesize that within fungi, differentiation and specialization exist among milRNAs based on their expression levels in regulating target gene functionalities. Those milRNAs exhibiting high expression levels are likely to play pivotal roles in fundamental biological processes, encompassing cellular proliferation, metabolic activities, and the maintenance of genomic stability, which are essential for the survival and development of the organism. Conversely, milRNAs with low expression profiles might be tasked with the modulation of specialized pathways, including the adaptation to environmental alterations, involvement in distinct developmental stages, or the fine-tuning of expression patterns of specific genes (Figure 5). Such specialization could potentially facilitate the maintenance of adaptability and stability of fungi in complex and variable environments.

The regulatory influence of milRNAs on themselves or their host plants occurs primarily through their target genes, rendering the roles of these target genes crucial for elucidating the biological significance of milRNAs. An analysis of GO biological functions revealed that the mRNA targets of CEmilRNAs across the three developmental stages of *FoHS2* can be categorized under 41 functional terms, including catalytic activities, metabolic processes, cellular processes, and biological regulation (Figure 3a). Furthermore, KEGG metabolic pathway analysis demonstrated that these target genes can be mapped to 20 metabolic pathways, such as translation, energy metabolism, amino acid metabolism, carbohydrate metabolism, the biosynthesis of other secondary metabolites, and the biosynthesis and metabolism of polysaccharides (Figure 3b). These findings imply that milRNAs exert extensive regulatory roles in the growth and development, material and energy metabolism, and environmental adaptation of the *FoHS2*.

In this study, the *Fo*_sp stage exhibited the highest abundance of milRNAs, exemplified by *FoHS2*-milR19, which targeted 5141 mRNAs (Figure 2a and Appendix A, and Appendix A), including those encoding histone acetyltransferases. These enzymes, alongside histone deacetylases (HDACs), regulate histone acetylation and deacetylation, pivotal epigenetic modifications enabling organisms to respond to environmental stimuli. Considering the significant impact of the overexpression and silencing of the histone deacetylase *MoRpd3* in *M. oryzae* on sporulation and pathogenicity [36,37], we speculated that *FoHS2*-milR19 is likely influencing spore production and pathogenic potential by regulating histone acetylation levels. Moreover, *FoHS2*-milR19 targets multiple methyltransferases and related genes, which play important roles in fungal growth, development, and pathogenicity. Notably, the m6A methyltransferase, *CpMTA1* [38], and the arginine methyltransferase, *PeRmtC* [39], play key roles in regulating these processes in plant pathogenic fungi. The deletion of *PeRmtC* impacts mycelial growth, spore production, and regulates processes related to cell wall integrity, environmental stress response, and pigment biosynthesis. Based on these observations, we inferred that *FoHS2*-milR19 may further regulate the proliferation, maturation, and virulence of spores through methylation modification pathways. Additionally, *FoHS2*-milR19 targeted genes encoding the serine/threonine protein kinase *Rim15* in *M. oryzae*, which is essential for saprophytic growth and suppression of plant defenses [40]. Our study also found *FoHS2*-milR19 acting on the developmental regulator *flbA*. Previous research had shown that *LqFlbA* in *Leptosphaeria konigii* plays a central role in fungal growth, development, and pathogenicity [41]. In summary, *FoHS2*-milR19 exhibits multifaceted regulatory roles within *FoHS2*, impacting not only its growth and development processes but also a multitude of genes associated with pathogenicity.

Intriguingly, some highly expressed CEmilRNAs in this study were predicted to have only a few target genes. For example, both *FoHS2*-milR1 and *FoHS2*-milR2 showed high expression level across the three growth stages but had relatively few predicted targets (Appendix A). A similar phenomenon has been reported in *Populus tomentosa*, where four highly expressed miRNAs (miR168, miR390, miR403, and miR479) did not have detectable specific targets [42]. However, it is noteworthy that previous studies have demonstrated that the homolog milRNA of *FoHS2*-milR2 in *Fol*, named *Fol*-milR1, can be translocated across kingdoms into the host plant and suppress host immunity by targeting host resistance genes, thereby avoiding the activation of plant defenses and promoting the pathogen’s infection of the host [22]. Additionally, the homologous milRNA of *FoHS2*-milR2, known as milR87 in *Foc*, targets a gene encoding a glycosyl hydrolase enzyme, which is a negative regulator of mycelium growth and virulence in *Foc*, responsible for activating plant defense responses [21]. Thus, we speculated that globally highly expressed milRNAs, such as *FoHS2*-milR2, may be involved in the interaction process between the pathogen and the host mainly through two main mechanisms. Firstly, they may promote pathogen infection by inhibiting the negative regulators of fungal growth and virulence. Secondly, they may be transferred across kingdoms into the host plant and target host disease resistance genes to suppress host immunity, thereby functioning akin to fungal pathogen effectors.

The fungal cell wall acts as a defensive shield, and maintaining its stability is vital for fungi to withstand environmental pressures. The primary constituents of this wall include chitin, cellulose, and deacetylated chitosan [43]. In filamentous fungi, the synthesis and degradation of chitin in the fungal cell wall are very important for fungal growth and development [44]. Previous studies had indicated that chitin synthase genes can perform multiple functions to regulate the growth of *Verticillium dahliae*, its stress responses, and its virulence towards the host [45]. In this study, the targets of milRNAs (*FoHS2*-milR19, *FoHS2*-milR41, *FoHS2*-milR53, etc.) encode eight chitin synthase genes (Appendix A). These milRNAs are implicated in the regulation of chitinase production and are hypothesized to play a crucial part in maintaining cellular structure stability, as well as in the ontogeny and development of *FoHS2*.

Furthermore, there is a close correlation between the pathogenicity of fungi and the CWDEs they secrete. For instance, pectinases in *Colletotrichum coccodes* [46] and *Peronophythora litchii* [47] play significant roles in their pathogenicity. The *FoHS2*-milR20 and *FoHS2*-milR29, which are commonly expressed across the three growth stages, target three pectinase genes (contig000006.606, contig000001.633, and contig000001.65) (Appendix A), and the spore stage-specific *FoHS2*-milR19 targets two pectate lyase genes (contig000006.116 and ontig000010.379) (Appendix A). We speculated that these milRNAs may selectively regulate the expression level of these pathogenicity-associated CWDE genes. Actually, previous studies have provided compelling evidence for the regulatory mechanisms controlling the expression of CWDE genes in pathogens under varying growth conditions. For example, the milRNA16 in the *V. mali* suppresses virulence genes expression during axenic culture but derepresses them during host plant interaction, thereby enhancing virulence and infection capability [48]. Additionally, the *FoHS2*-milR20 and the *FoHS2*-milR19 can target several glycosyl hydrolase genes (Appendix A). In *F. oxysporum*, the glycoside hydrolase gene *FoEG1* contributes to the pathogen’s virulence based on its enzymatic activity [49]. Similarly, previous studies found that *PEC032*, encoding an α-mannosidase in *Podosphaera xanthii*, affects fungal growth and development and leads to the accumulation of hydrogen peroxide in host plants, which is crucial for the pathogenicity and survival of fungi during plant colonization [50]. The α-galactosidase *VdGAL4* from *Verticillium dahliae* is known to activate plant immune responses and plays a significant role in spore formation and hyphal penetration [51]. Therefore, we speculated that milRNAs may regulate the pathogenicity of *FoHS2* by modulating the expression levels of extracellular hydrolase.

During the interaction between pathogenic fungi and plants, secondary metabolites serve as key virulence factors, involving complex signal transduction pathways and interactions with environmental factors in their production and regulation [52]. Mycotoxins, as a significant component of secondary metabolites, are closely associated with the pathogenicity of pathogens [53]. For instance, the deletion of secondary metabolite toxin synthesis genes in *Fusarium* species not only results in the loss of the ability to produce these toxins but also significantly reduces their pathogenicity towards host plants [54]. Treatment with exogenous crude fusaric acid toxin significantly inhibits the germination and growth of *Medicago sativa* seeds and exhibits strong pathogenicity to *M.Sativa* seedlings [55]. In this study, we identified spore-specific milRNAs that regulate the expression of 68 target mRNAs (including contig000002.1035, contig000002.1466, contig000002.1664, etc.) annotated as being involved in secondary metabolite biosynthesis pathways (Additional file: Appendix A). Furthermore, it has been reported that milRNAs in *Fon* can regulate the expression of toxin-related genes [56]. Based on these results, we hypothesize that the milRNAs of *FoHS2* could indirectly influence pathogenicity by finely tuning the expression levels of mRNAs associated with secondary metabolite synthesis.

The importance of ATP-binding cassette (ABC) transporters and mitogen-activated protein kinases (MAPK) as essential components of the signaling pathways for *FoHS2* to achieve its full pathogenicity has been well established in various studies [57,58,59]. In this study, *FoHS2*-milR20 and multiple SEmilRNAs (csi-miR167c-3p, ghr-miR2948-5p, *FoHS2*-milR19, etc.) from the spore stage targeted three and twelve genes annotated to the MAPK signaling pathway, respectively (Appendix A). Additionally, seven target mRNAs (contig000002.306, contig000003.1175, contig000001.1536, etc.) of *FoHS2*-milR19 encode ABC transporters or related proteins (Appendix A). Studies have shown that ABC transporter *CDR4* in *N. crassa* is responsible for transporting toxins within host plant cells during fungal infection and enhancing its own drug resistance [60,61], and the plant pathogen *F. graminearum* exhibits reduced virulence to wheat following the knockout of genes encoding ABC transporters *FgABC1* and *FgABCC9* [62,63]. Research on MAPK has also demonstrated its connection with pathogenicity. For instance, the *F. graminearum Δgpmk1*, *Δmgv1*, and *Δfghog1* triple mutant exhibited severe growth defects and lacked pathogenicity, showing impaired infection cushion formation and deoxynivalenol (DON) production [64]. Additionally, studies have revealed that *BcMkk1* in *Botrytis cinerea* negatively modulates virulence by suppressing oxalic acid (OA) biosynthesis [65]. Based on this, it is inferred that milRNAs in *FoHS2* may regulate its own virulence by influencing ABC transporters and the MAPK signaling pathway.

Overall, this study found that the *FoHS2* exhibits a diverse expression of milRNAs from the spore stage to the mycelium stage, with the most extensive SEmilRNA occurring during the spore stage. This may indicate that the spore stage is a critical period for milRNA regulation of fungal growth, development, and pathogenicity towards the host apple by targeting multiple key genes. Therefore, thoroughly exploring the milRNA resources of *F. oxysporum* and accurately identifying and analyzing their targets will not only help us fully understand the specific roles of milRNAs in fungal proliferation, morphogenesis, and virulence but may also provide important molecular targets and theoretical foundations for the development of new and efficient disease control strategies.

## 5. Conclusions

This study identified one hundred and twenty-seven, seventy, and fifty-nine milRNAs in the three growth stages of the *FoHS2*, with ninety-four, seventeen, and three SEmilRNAs in each stage, respectively. The milRNAs across the three growth stages share similar structural characteristics, but their expression profiles show significant differences. Notably, *FoHS2*-milR19 targets the greatest number of mRNAs and has the richest annotated functions, including histone acetyltransferases, methyltransferases, and cell wall-degrading enzymes, which regulate the growth, development, and pathogenicity of the pathogen itself. Therefore, the milRNAs of the *FoHS2* may regulate its proliferation, morphogenesis, and virulence through stage-specific and differential expression, with the spore stage being a critical period for milRNA regulation of these functions in *FoHS2*.

## Figures and Tables

**Figure 1 jof-10-00883-f001:**
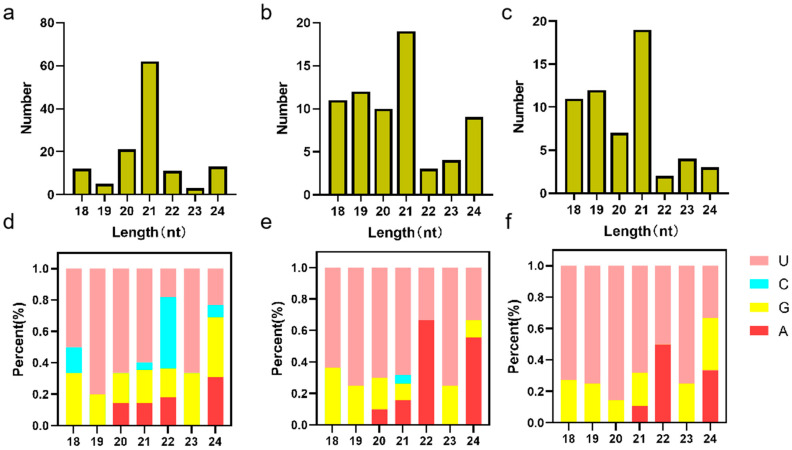
Structural characteristics of milRNAs in *FoHS2* at different growth stages. (**a**–**c**) Length and size distribution of spore (*Fo*_sp), germination (*Fo*_gs), and mycelium (*Fo*_myc). (**d**–**f**) Nucleotide bias distribution of spore (*Fo*_sp), germination (*Fo*_gs), and mycelium (*Fo*_myc).

**Figure 2 jof-10-00883-f002:**
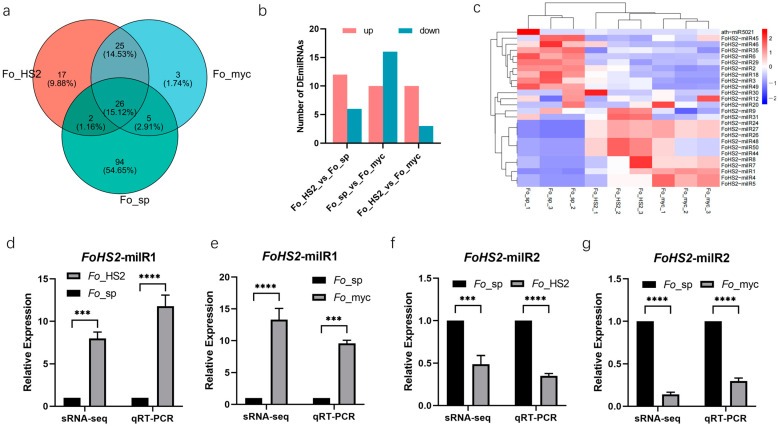
Expression pattern of milRNAs in *FoHS2* across developmental stages. (**a**) Venn diagram of milRNAs in different stages. (**b**) Number of differentially expressed milRNAs (DEmilRNAs) in different stages. (**c**) Heatmap of DEmilRNAs in *Fo*_sp, *Fo*_gs, and *Fo*_myc. (**d**–**g**) Verification of milRNA expression via qRT-PCR and statistical analysis. Significant differences were determined by two-way ANOVA. ***: *p* < 0.001, ****: *p* < 0.0001, respectively. Error bars indicate SEM.

**Figure 3 jof-10-00883-f003:**
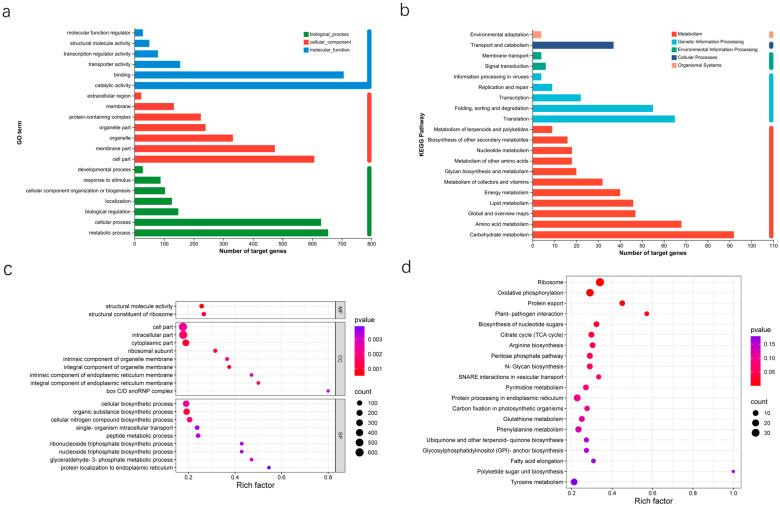
Functional annotation of targets of milRNAs in *FoHS2.* (**a**) GO annotations analysis. (**b**) KEGG annotation analysis. (**c**) GO enrichment analysis. (**d**) KEGG pathway enrichment analysis.

**Figure 4 jof-10-00883-f004:**
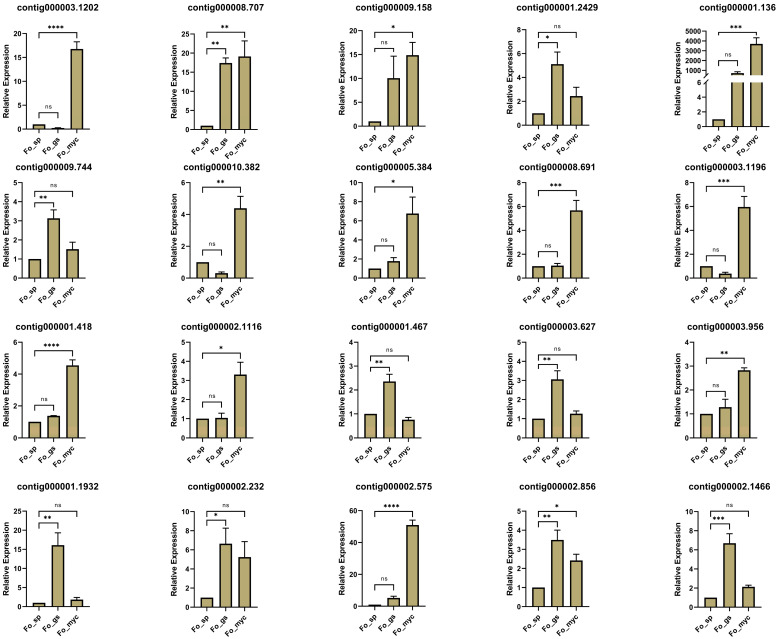
Expression levels of target genes regulated by *FoHS2*-milR19, derived from transcriptome sequencing data. ns: not significant; *: *p* < 0.05; **: *p* < 0.01; ***: *p* < 0.001, ****: *p* < 0.0001.

**Figure 5 jof-10-00883-f005:**
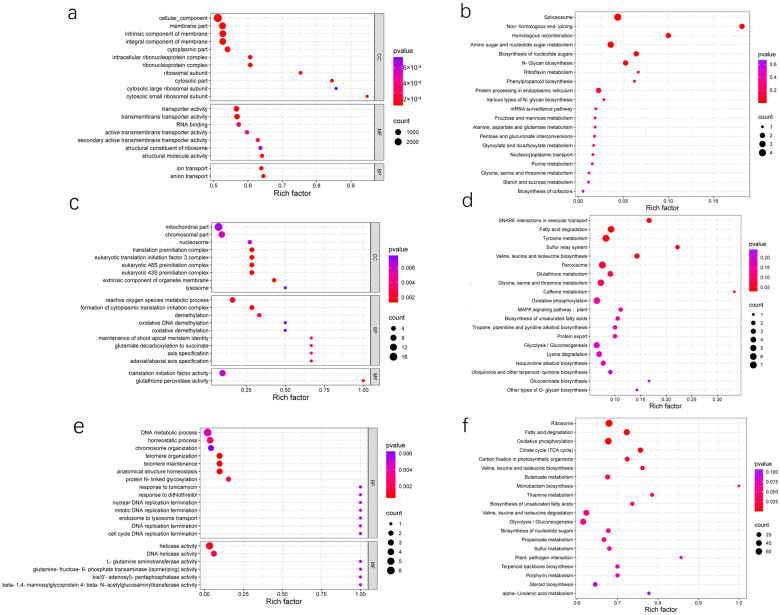
Enrichment analysis of targets for stage-specific milRNAs in *FoHS2*. (**a**) GO enrichment analysis of SEmilRNAs in *Fo*_sp. (**b**) KEGG enrichment analysis of SEmilRNAs in *Fo*_sp. (**c**) GO enrichment analysis of SEmilRNAs in *Fo*_gs. (**d**) KEGG enrichment analysis of SEmilRNAs in *Fo*_gs. (**e**) GO enrichment analysis of SEmilRNAs in *Fo*_myc. (**f**) KEGG enrichment analysis of SEmilRNAs in *Fo*_myc. (*p* < 0.05).

**Table 1 jof-10-00883-t001:** Data summary of three sRNA-seq from *FoHS2*.

Sample	Clean Reads	Useful Reads	Ratio
*Fo*_sp_1	6,688,780	6,453,879	96.49%
*Fo*_sp_2	7,112,552	6,674,966	93.85%
*Fo*_sp_3	4,875,547	4,706,746	96.54%
*Fo*_gs_1	6,721,992	6,307,945	93.84%
*Fo*_gs_2	6,506,702	6,067,401	93.25%
*Fo*_gs_3	4,904,334	4,657,173	94.96%
*Fo*_myc_1	3,115,454	3,062,599	98.30%
*Fo*_myc_2	3,181,705	3,133,593	98.49%
*Fo*_myc_3	3,282,875	3,234,971	98.54%

**Table 2 jof-10-00883-t002:** Target gene annotation information of *FoHS2*-milR19.

milRNA Name	Target Transcript ID	Target Gene Description
*FoHS2*-milR19	contig000015.104	RKK08350.1 Histone acetyltransferase GCN5
*FoHS2*-milR19	contig000012.106	RKK08350.1 Histone acetyltransferase GCN5
*FoHS2*-milR19	contig000006.442	EMT66086.1 Histone acetyltransferase ESA1
*FoHS2*-milR19	contig000003.1109	EXM07738.1 Histone-lysine N-methyltransferase
*FoHS2*-milR19	contig000001.2042	EMT66938.1 Phosphoethanolamine N-methyltransferase
*FoHS2*-milR19	contig000004.886	EXK92416.1 Methyltransferase
*FoHS2*-milR19	contig000004.1301	ENH63214.1 Putative methyltransferase BUD23
*FoHS2*-milR19	contig000005.235	EXL95464.1 Histone-lysine N-methyltransferase ASH1L
*FoHS2*-milR19	contig000001.473	EMT70982.1 Histone-lysine N-methyltransferase
*FoHS2*-milR19	contig000003.1030	XP_018234307.1 Serine/threonine protein kinase
*FoHS2*-milR19	contig000002.1490	EMT61253.1 Serine/threonine protein kinase sid1
*FoHS2*-milR19	contig000001.306	EMT70782.1 Serine/threonine protein kinase SRPK3
*FoHS2*-milR19	contig000004.1044	EMT60585.1 Developmental regulator flbA
*FoHS2*-milR19	contig000013.90	RKK89124.1 Developmental regulator flbA
*FoHS2*-milR19	contig000007.796	RKK65984.1 Developmental regulator flbA
*FoHS2*-milR19	contig000001.930	XP_018240063.1 Chitin synthase 3
*FoHS2*-milR19	contig000001.1709	XP_018238709.1 Chitin synthase
*FoHS2*-milR19	contig000001.633	EMT63622.1 Putative pectate lyase F
*FoHS2*-milR19	contig000006.116	XP_018250110.1 Pectate lyase plyB
*FoHS2*-milR19	ontig000010.379	EXL97925.1 Pectate lyase
*FoHS2*-milR19	contig000007.142	EMT67745.1 Putative pectate lyase F

## Data Availability

The original contributions presented in this study are included in the article/Appendix A. Further inquiries can be directed to the corresponding authors.

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
