# Peer review of "Global Analysis of microRNA-like RNAs Reveals Differential Regulation of Pathogenicity and Development in Fusarium oxysporum HS2 Causing Apple Replant Disease"

_jof, 2024, doi:10.3390/jof10120883_

Round 1
Reviewer 1 Report
The paper entitled 'Global analysis of microRNA-like RNAs reveals differential regulation of pathogenicity and development in Fusarium oxysporum HS2 causing apple replant disease' is devoted to the analysis of micro-RNA-like RNAs profiles at different stages of fungal growth. Indeed, microRNAs are one of the most significant tools of regulation of gene expression both in pathogen and host plant. However, their action in Fungi, including plant-pathogenic Fusarium species, have been little investigated. So, the paper looks timely and scientifically sound. The manuscript is very well written and comprehensively describes the results obtained.
Therefore, the paper can be accepted for publication in JoF after some minor improvements (see below).
1. The authors should verify the text in terms of reference design ([..]) and mystakes (for instance, missing words).
2. It would be interesting to have a couple of phrases in Introduction about miRNA formation mechanisms in eukaryotic organisms, including Fungi.
3. More information about the FoHS2 strain (origin, host plant, year) is needed.
4. How the strong preference for uracil as the initial base of milRNA can be explained?
5. In Discussion, the authors can mention whether similar experiments (analyzing microRNA profiles at different developmental stages of any fungal species) were performed?
Line 34: ....18-24 nucleotides, (MISSING) in various biological processes. - improve
Line 112: It is not clear, whether QIAseq miRNA Library kit was used for RNA extraction or just for libraries preparation.
Line 117-118: RIN numbers for the samples can be presented in a separate table or in the text.
Author Response
Thank you very much for your insightful comments and suggestions. Please find our detailed point-by-point responses below, and note that the corresponding revisions and corrections have been highlighted and incorporated into the re-submitted manuscript.
Comments1. The authors should verify the text in terms of reference design ([..]) and mystakes (for instance, missing words).
Response1:
Thank you for your valuable comments. We have carefully examined the text on the format of the reference and the possible spelling errors.
Comments2. It would be interesting to have a couple of phrases in Introduction about miRNA formation mechanisms in eukaryotic organisms, including Fungi.
Response2:
Thank you very much for your suggestion. We have added a description of the mechanism of fungal milRNA formation, see lines 46-52 of the manuscript.
Comments3. More information about the FoHS2 strain (origin, host plant, year) is needed.
Response3:
Thank you for highlighting the importance of detailing the strain background, particularly for species of FoHS2. We have described it in the manuscript, see lines 101-103.
Comments4. How the strong preference for uracil as the initial base of milRNA can be explained?
Response4:
Thank you very much for your comments. This preference may be related to the selectivity of Argonaute protein, which shows a strong preference for 5'-uracil when selecting the guide chain of miRNA double strand. This selectivity may be related to the maturation and function of miRNAs, especially during the loading of the guide strand into the RNA-induced silencing complex (RISC). Moreover, it is similar in other fungal miRNA studies, all of which show the first base preference for uracil. We explained it in the manuscript, see lines 198-201.
Comments5. In Discussion, the authors can mention whether similar experiments (analyzing microRNA profiles at different developmental stages of any fungal species) were performed?
Response5:
Thank you for your question. We describe this part in the discussion section, see lines 358-364.
Detail comment
Comments1 Line 34: ....18-24 nucleotides, (MISSING) in various biological processes. - improve
Response1:
Okay, we 've made changes, see line 35.
Comments2 Line 112: It is not clear, whether QIAseq miRNA Library kit was used for RNA extraction or just for libraries preparation.
Response2:
Thank you for your query regarding the use of the QIAseq miRNA Library Kit in our study. We have made adjustments, see lines 125-127.
Comments3 Line 117-118: RIN numbers for the samples can be presented in a separate table or in the text.
Response3:
Thank you for your suggestion to present the RIN numbers for the samples in a separate table or in the text. We have taken your advice into account and have made the following revisions to our manuscript:
We have included the RIN values for the samples in Table S1-3, as per your recommendation. This allows readers to easily access and review the RNA integrity data alongside other supplementary information.
Upon further review, we realized that the quality control measure used was actually the RNA Quality Number (RQN) instead of RIN. We have corrected this oversight and have now presented the RQN values in Table S1-3. We have also updated the methods section to accurately reflect the use of RQN for quality control purposes, ensuring that the description aligns with the data presented in the supplementary table. See line 130.
Reviewer 2 Report
The findings of this research are significant for the scientific community and help clarify the mechanisms of fungal development, especially regarding species of the genus Fusarium due to its extensive variety of races.
Describing the strain's background is crucial, especially regarding F. oxysporum species. This includes important details such as the source of isolation, the host organism, the date the isolate was collected, and the identification methods used.
Line 97: what conditions of light?
Line 99: pelle?
Define how to determine if a spore is germinated or not germinated.
Lines 129 and 135: Access or link is not enabled.
Author Response
Thank you very much for your insightful comments and suggestions. Please find our detailed point-by-point responses below, and note that the corresponding revisions and corrections have been highlighted and incorporated into the re-submitted manuscript.
Detail comment
Comments1: Describing the strain's background is crucial, especially regarding F. oxysporum species. This includes important details such as the source of isolation, the host organism, the date the isolate was collected, and the identification methods used.
Response1:
Thank you for highlighting the importance of detailing the strain background, particularly for species of F. oxysporum. In response to your request for further clarification, we have provided the following comprehensive information regarding our strain, FoHS2:
Source of Isolation: The strain FoHS2 was isolated from the roots of apple trees in Hengshui, Hebei Province.
Host Organism: The host organism for this strain is Malus domestica, commonly known as the apple tree.
Collection Date: The isolation of the strain was conducted in the year 2011.
We have described it in the manuscript, see lines 101-103.
Comments2: Line 97: what conditions of light?
Response2:
Thank you for your insightful question regarding the light conditions mentioned in our manuscript on Line 97. Our dark incubation refers to the condition of avoiding all sources of light, including both natural and artificial light.
Comments3: Line 99: pelle?
Define how to determine if a spore is germinated or not germinated.
Response3:
Thank you for your question. Regarding the production of PDA (Potato Dextrose Agar), we use peeled potatoes.
Thank you for your inquiry regarding the criteria used to determine the germination status of spores in our study. We have addressed this in the manuscript, as indicated in lines 115-117, and Figure S1 provides a visual representation of the three-stage morphology.
Comments4: Lines 129 and 135: Access or link is not enabled.
Response4:
Thank you for your advice. Regarding your comments on Lines 129 and 135 about 'Access or link is not enabled,' we have promptly reviewed and ensured that all links are active and accessible.